# Implementation of a Virtual Reality Based Digital-Twin Robotic Minimally Invasive Surgery Simulator

**DOI:** 10.3390/bioengineering10111302

**Published:** 2023-11-09

**Authors:** Xiaoyu Cai, Zijun Wang, Shijie Li, Junjun Pan, Chengli Li, Yonghang Tai

**Affiliations:** 1School of Physics and Electronic Information, Yunnan Normal University, Kunming 650000, China; caixiaoyu@user.ynnu.edu.cn (X.C.); zijunwang@ynnu.edu.cn (Z.W.); lishijie@ynnu.edu.cn (S.L.); 2State Key Laboratory of Virtual Reality Technology and Systems, Beihang University, Beijing 100191, China; pan_junjun@buaa.edu.cn

**Keywords:** robotic minimally invasive surgery, digital twin, virtual reality, training simulator, remote center of motion, entropy

## Abstract

The rapid development of computers and robots has seen robotic minimally invasive surgery (RMIS) gradually enter the public’s vision. RMIS can effectively eliminate the hand vibrations of surgeons and further reduce wounds and bleeding. However, suitable RMIS and virtual reality-based digital-twin surgery trainers are still in the early stages of development. Extensive training is required for surgeons to adapt to different operating modes compared to traditional MIS. A virtual-reality-based digital-twin robotic minimally invasive surgery (VRDT-RMIS) simulator was developed in this study, and its effectiveness was introduced. Twenty-five volunteers were divided into two groups for the experiment, the Expert Group and the Novice Group. The use of the VRDT-RMIS simulator for face, content, and structural validation training, including the peg transfer module and the soft tissue cutting module, was evaluated. Through subjective and objective evaluations, the potential roles of vision and haptics in robot surgery training were explored. The simulator can effectively distinguish surgical skill proficiency between experts and novices.

## 1. Introduction

Over the past 30 years, minimally invasive surgery (MIS) has significantly impacted surgical techniques across various specialties and has nearly supplanted traditional open surgery. Nonetheless, MIS presents two primary drawbacks: an extended learning curve for minimally invasive procedures and the challenge of addressing the surgeon’s hand tremors, which can significantly impact intraoperative bleeding and postoperative recovery [1,2]. Since the birth of the Automated Endoscopic System for Optimal Positioning (AESOP) as the first FDA-certified auxiliary surgical robot in 1994, various types of surgical robot systems have sprung up like mushrooms. The most renowned and successful platform is the da Vinci Surgical Robot developed by Intuitive Surgical^®^ Inc (Sunnyvale, CA, USA). The system was initially used for remote military minimally invasive surgery and was successfully commercialized to surgical hospitals. The main innovation of the da Vinci system lies in the integrated multi-degree-of-freedom surgical robotic arm, which has 3D vision and a 5 mm scope endoscope at the same time [3]. However, most RMISs currently lack tactile feedback, which impairs surgeons’ ability to perceive tissue texture, tension, and 3D depth [4]. Consequently, most of the current research hotspots on surgical robots tend to enable them to have the ability to provide tactile feedback [5] and shorten the learning curve for junior surgeons [6].

The control method for RMIS differs from traditional surgical instruments. A distinct approach is employed where the leader and responder components are separated [7,8]. In other words, the secondary robot is not directly controlled by the surgeon’s hand but is manipulated by the surgeon through motion capture using the leader console. The framework is depicted in Figure 1.

Given that most surgeons are accustomed to traditional minimally invasive surgery (MIS) methods, there is a significant adjustment period required for them to transition from conventional techniques to the current methods employed in surgical robot operations. Therefore, the learning curve for robotic minimally invasive surgery (RMIS) becomes notably steep. To alleviate the learning curve of controlling RMIS, researchers have developed incorporating advanced technologies such as three-dimensional displays, haptic feedback, and virtual reality/augmented reality (VR/AR) [9,10,11,12,13,14]. Currently, there is a substantial shortage of cadaveric resources in medical schools, which presents a major hindrance for surgeons seeking to undergo RMIS training. Therefore, VR/AR technology offers a promising solution to help surgeons enhance their RMIS skills without the need for cadavers [15,16].

This paper proposes a robotic minimally invasive surgical simulator based on VR digital twins (VRDT-RMIS) and verifies the surgical skill training effect of the simulator through two module experiments.

Our contributions:A set of laparoscopic surgical robot virtual reality simulators, encompassing both software and hardware design, has been devised and validated;The simulator has been evaluated using two modules: peg transfer and soft tissue cutting;The effectiveness of simulator training has been quantitatively assessed through the utilization of the entropy method.

The article is organized as follows: Section 2 presents the design of the robot, including the remote center of motion (RCM) algorithm for robot control and the hardware–software framework. Section 3 primarily focuses on introducing the experimental design, including participants, experimental tasks, data collection, and data analysis methods. Section 4 integrates the aforementioned evaluation methods to assess face and content validation and construct validation. It also provides a detailed description of calculating the weight of different metrics in surgical skills using the entropy method based on expert surgical proficiency. This serves as a reference for evaluating the training outcomes of novice doctors on the surgical robot simulator in the future. The results of data analysis are presented in Section 5, where the significant differences in performance between experts and novices in different tasks are discussed. This affirms the positive significance of this simulator in training doctors to operate robotic minimally invasive surgery robots.

## 2. Design of the VRDT-RMIS

### 2.1. Remote Center of Motion Algorithm

In traditional MIS, surgeons receive thorough training in endoscopic procedures to ensure that their surgical instruments operate within the confines of a trocar. This practice significantly minimizes the risk of unintentional injuries occurring outside the intended target area. However, due to the dissimilarity between robotic inverse kinematic (IK) motion and the natural hand motions of surgeons, it is imperative to incorporate a remote center of motion (RCM) algorithm into the robot control system. This addition enables robotic MIS to replicate the surgeon’s actions effectively, thereby reducing patient wounds and minimizing intraoperative bleeding.

In recent years, as surgical robot technology has advanced, the RCM algorithm has undergone significant refinement. In 2013, Nastaran and colleagues [17] addressed a visual task involving a manipulator with six degrees of freedom, specifically one holding an endoscopic camera. An inverse kinematic control law was successfully derived, allowing the manipulator to perform a visual task while adhering to the RCM constraint. However, it should be noted that their approach only guaranteed local convergence, and external factors such as the patient’s involuntary movements, for example, heartbeats and breathing, were not considered. In 2019, a novel design based on artificial neural network architecture was introduced by Ahmed and colleagues [18]. Their work demonstrated the feasibility and effectiveness of utilizing neural networks in the context of RMIS. In 2020, an approach combining model-based and model-free methods for calibrating and controlling robots in terms of their forward kinematics and inverse kinematics was proposed by Alireza and colleagues [19]. The aforementioned approaches are unable to provide real-time control of the robot with haptic feedback, potentially leading to suboptimal surgical outcomes. In 2021, a platform was designed by Oliva Wilz et al. [20] that permits a 3-DOF haptic device to control a 6-DOF manipulator. A novel design of concurrent inverse kinematics solvers that combines classical Jacobian inverse kinematics and optimization-based methods was presented by Jacinto and colleagues [21] in 2023.

Remote Center of Motion Constraint Task

The remote center of motion (RCM) constraint is defined based on the kinematic distance between the trocar point and the tool axis [22]. The locations of the joints before and after the RCM point are defined as $L_{before}\in R^{3\times 1}$ and $L_{after}\in R^{3\times 1}$. The point near the trocar point is defined as $L_{rcm}\in R^{3}$, allowing us to obtain (1).
(1)$$L_{rcm}=L_{pre}+L_{r}^{T}\hat{L_{s}}\hat{L_{s}},$$
where ${\hat{L}}_{s}=\frac{L_{post}-L_{pre}}{||L_{post}-L_{pre}||}$ means the direction of the surgical instrument axis, and $L_{r}=L_{trocar}-L_{pre}$ represents the disparity in position between the trocar point and its location before the RCM point. In addition, the vector to the trocar point, $L_{trocar},$ from its nearest point on the instrument axis, $L_{rcm},$ is denoted as $L_{e}=L_{trocar}-L_{rcm}$. The above definition is shown in Figure 2.

As differentiating $L_{rcm}$ with respect to the manipulator joints, $q\in R^{n\times 1}$, the RCM constraint task in the Jacobian matrix, $J_{rcm_{\left(q\right)}}\in R^{1\times n},$ could be calculated as
(2)$$J_{rm\left(q\right)}=L_{e}^{T}\frac{\delta L_{tcm}}{\delta q}=L_{e}^{T}\left[\left(I_{3}-\hat{L_{s}}{\hat{L_{s}}}^{T}\right)J_{before_{\left(q\right)}}+\left(\hat{L_{s}}L_{r}^{T}+L_{r}^{T}\hat{L_{s}}I_{3}\right)\frac{\delta \hat{L_{s}}}{\delta q}\right],$$
with
(3)$$\frac{\delta {\hat{L}}_{s}}{\delta q}=\frac{1}{||L_{s}||}\left(I_{3}-{\hat{L}}_{s}{\hat{L}}_{s}^{T}\right)\left(J_{before_{\left(q\right)}}-J_{after_{\left(q\right)}}\right),$$
where $J_{before}\in R^{3\times n}$ and $J_{after}\in R^{3\times n}$ are the configuration-dependent analytical Jacobian matrices of the immediate joints before and after the RCM.

The residual of the RCM constraint task, denoted as $e_{rcm},$ is defined as the minimum distance to the trocar point, and can be calculated as follows:(4)$$e_{rcm}=||L_{e}||=||L_{trocar}-L_{rcm}||,$$

The residual of the RCM task, $e_{rcm},$ and the Jacobian representation of the RCM constraint, $J_{rcm}$, will be used to calculate the hierarchical constraint inverse kinematic formulation.

Inverse Jacobian Inverse Kinematic Solver

The classical Jacobian method is a widely used approach for solving inverse kinematic problems, valued for its simplicity and low computation cost. This method is rooted in the concept of differential kinematics, expressed as $\dot{x}=J_{\left(a\right)}\dot{q},$ where $\dot{x}\in R^{\hat{m}\times 1}$ represents the task space vector, q is the joint vector of the manipulator, and $J_{\left(q\right)}\in R^{n\times m}$ is the configuration-dependent Jacobian matrix task, connecting joint velocities with task space velocities. When given a desired task space velocity, ${\dot{x}}_{des}$, the inverse kinematic problem will transform into the task of finding a solution of $q^{*}$, as depicted in Equation (5).
(5)$${\dot{q}}^{*}=\arg\;\underset{\dot{q}}{\min}||J_{\left(q\right)}\dot{q}-{\dot{x}}_{des}|{|}^{2},$$

A least squares solution can be obtained by utilizing the Moore–Penrose pseudoinverse of $J_{\left(q\right)}$, as illustrated in Equation (6).
(6)$${\dot{q}}^{*}=J_{\left(q\right)}^{†}{\dot{x}}_{des}+Pz,$$
where $J^{†}=J^{T}{\left(JJ^{T}\right)}^{-1}$ and P presents the orthogonal projection operator in the null space of J, while z is an arbitrary task velocity vector which cannot disturb task ${\dot{x}}_{des}$.

Following this, if k tasks with different priorities are assigned, the inverse kinematic problem can be solved for each task in the null space by the previous one, thus creating a hierarchy between tasks. The recursive formula is defined in (9) [23].
(7)$$q_{i}^{*}=q_{i-1}^{*}+P_{i-1}^{A}J_{i}^{†}e_{i},$$
where i=1,…,k, and $P_{i}^{A}$ denotes the projector in the null space of the augmented Jacobian matrix $J_{i}^{A}=\left(J_{1},\ldots ,J_{i}\right)$.

Considering an RCM constraint task, the desired joint velocities will be given as
(8)$$\dot{q}=J_{rcm}^{†}e_{rcm}+\left(I_{n}-J_{rcm}^{†}J_{rcm}\right),$$

### 2.2. Hardware

#### 2.2.1. Leader Part

The digital-twin robots are primarily controlled using 3DSystem Touch devices (3D Systems^®^, Rock Hill, SC, USA). These devices are capable of refreshing at 1000 Hz, which is on par with the tactile sensation of human hands. The data processing part is a computing station which is equipped with an Ubuntu 20.04 LTS system and an Nvidia GTX3060 graphics card (Nvidia Santa^®^ Clara, CA, USA). There are virtual reality (VR) glasses, named Pimax 8K X (Pimax^®^, Shanghai, China), which provide a 200° field of view (FOV).

The hardware components of the leader part of the system are depicted in Figure 3 and their details are listed in Table 1.

#### 2.2.2. Responder Part

To facilitate the manipulation of target objects at a distance, two responder robots equipped with end-effectors are utilized as remote extensions of the surgeon’s hands. A 1920 × 1080 endoscope serves as the video input device, delivering a high-quality view for the operators. The components of the responder part are illustrated in Figure 4 and are detailed in Table 2.

### 2.3. Software

A system was designed to enable the real-time control of the digital-twin robots, as illustrated in Figure 5. This system is built upon platforms known as Unity3D [24] (2021.3.30f1 LTS) and RoboDK v5.6.3. [25,26].

To ensure an effective integration of hardware and software, a comprehensive framework has been meticulously designed, as depicted in Figure 6. It combines bottom-up mechanisms through the hardware layer, the robots’ digital-twin communication layer, and the visual and haptic rendering layer. The Universal Robots and 3DSystem Touch devices are seamlessly integrated into the hardware layer, where they gather data from the operator’s movements. Once the motion data are collected within a 2 ms interval, it is packaged and transmitted to the digital-twin communication layer for noise filtering and processing through the remote center of motion (RCM) inverse kinematic algorithm. Subsequent to the data processing phase, Unity3D extracts digital-twin (DT) data and translates them into visual and haptic feedback data for the leader part. This comprehensive framework ensures smooth interaction and communication between the hardware and software components.

## 3. Experimental Design

### 3.1. Participants

A total of 25 doctors, comprising 10 experts and 15 novices, were invited to participate in this experiment. The selection criteria for participants were based on their robotic minimally invasive surgery experience, with novices having no more than 3 years of experience, while experts had a minimum of 6 years of experience. Detailed participant information is provided in Table 3.

Due to a low proficiency in RMIS operation, it takes novices more than 2 weeks to practice before their actual experiments. All of the participants should be familiar with the operating rules and training tasks before conducting experiments to ensure the validity of the data. The period of the whole experiment is exhibited in Figure 7.

### 3.2. Tasks

At the outset of the experiment, a designated experimental recorder is tasked with recording the hand, head, and instrument movements of all participants during the experiment. These recordings will serve as the raw data for the subsequent assessment of their surgical skills. The experiment primarily consists of two key tasks [27,28]: peg transfer and soft tissue cutting. Following the completion of one training mode, participants are provided with a 30 min break before the next training mode to ensure that their experimental data are not influenced by subjective physical fatigue.

To enhance the accuracy of evaluating the effectiveness of the various training modes in the experiment, all operators will receive detailed information regarding the scoring criteria for the operations and the specific operational requirements for each training module before the experiment commences. In the peg transfer module, the operators should employ the left surgical instrument to elevate the small pegs positioned on the left side and subsequently use the right surgical clamp to grasp and transfer the pegs to the columns on the right. If a small object falls during the operation, it will be counted as the number of drops, but the operator can use the instruments to pick up the object and continue with the previous operation. In the soft tissue cutting mode, the operators must utilize the left surgical clamp to grasp and gently tug on the soft tissue, followed by using the right surgical knife to precisely remove the specified soft tissue entirely. The duration, operation path, and cutting frequency used to complete the entire step will be recorded.

These two training modules are shown in Figure 8. Through the above two modules, the surgical skills of the experimental participants can be recorded, and their surgical skills will be quantitatively reflected.

### 3.3. Data Collection

Because the simulator cannot automatically record all the data required for our experiment, the recorder will manually document relevant information during the operation. This includes the number of objects dropped in the peg transfer module and the cutting frequency in the soft tissue cutting module. A comprehensive surgical skill assessment will be conducted after merging the experimental data. A comparison will be made between the Novice Group and the Expert Group. Common evaluation criteria for both modules include the total experiment time (*T*) and the length of movement of surgical instruments (*ML*) for both the left and right instruments. Additionally, specific tasks have unique parameters to consider. In the peg transfer Module, the evaluation will consider the number of peg drops (*ND*), whereas for the soft tissue cutting task, performance will be assessed using the cutting frequency (*FC*).

This paper assesses the pros and cons of different modules by employing face and content validation through the use of questionnaires. The questionnaire employs a 5-point Likert scale to assess the visual and tactile experiences of the two training methods, with ratings ranging from 1 (Poor) to 5 (Excellent) [29]. The questionnaire, presented in Table 4, was completed by an expert team to provide a more precise evaluation of the simulator.

### 3.4. Data Analysis

For face and content validation, the average values of subjective questions were calculated using descriptive statistical methods. Prior to statistical analysis, a reliability analysis was performed on the questionnaires, with a Cronbach’s alpha coefficient exceeding 0.7 considered as a credible measure.

In terms of construct validation, we compared the experimental data from the Novice Group and the Expert Group using an independent sample *t*-test. The simulator’s impact is considered significant when the *p*-value is less than 0.05.

We utilize the entropy method to determine the weighting for each evaluation item within the modules. To mitigate the impact of physical quantities, we employ a range-based method, followed by the calculation of the coefficient of variance for each evaluation item, resulting in their respective index weights. This paper applies the entropy method to establish the weight of each evaluation item within various modules. Initially, the scope method is employed to standardize each evaluation item by removing dimensionality, thereby eliminating the influence of physical quantities on the data. Subsequently, the coefficient of variance for each evaluation item is calculated to determine its index weight. The score function for each training module is then used to compute the experimental scores for both the Novice Group and the Expert Group. Finally, a block diagram is employed to compare the specific performance of these two groups. The motion trajectory diagram of the surgical instruments will be used to reveal the proficiency of the two groups. A statistical analysis software (SciPy 1.8.1 Python 3.9.5, based on miniconda) was used for the whole analysis process.

## 4. Evaluation

### 4.1. Face and Content Validation

To assess the visual and haptic perception of various modules, we designed a subjective questionnaire using a 5-point Likert scale, consisting of five questions. The questionnaire data from Expert Group doctors after the experiment are presented in Table 5.

The reliability analysis of the statistical questionnaire yielded a total Cronbach’s alpha coefficient exceeding 0.7. Additionally, the data from each group exhibited a roughly normal distribution, as confirmed by the Shapiro–Wilk test.

### 4.2. Construct Validation

When the operator is training, the virtual operation simulator will automatically record the movement of the instrument and the completion time of the experiment.

Detailed data are provided in Table 6, illustrating that the Expert Group exhibits significantly higher proficiency compared to the Novice Group, as observed in both the figure and the table. The *p*-value results, displayed in Table 7, indicate the statistical significance of the differences between the Novice Group and the Expert Group.

### 4.3. The Entropy Method

#### 4.3.1. The Steps of the Entropy Method Steps

When selecting n experts and m evaluation items (i=1, 2…, n;j=1, 2, …, m), it is necessary to standardize the measurements of various indicators before calculating comprehensive scores. This standardization process involves converting absolute indicator values into relative values, addressing the differences in measurement scales. Furthermore, it is important to note that the interpretation of positive and negative indicators differs—higher values are better for positive indicators, while lower values are preferable for negative ones. Therefore, we employ distinct standardization algorithms for high and low indicators to account for these differences. The standardization process follows these specific methods:

Positive index:(9)$$x_{ij}^{\prime }=\frac{x_{ij}-min\left\{x_{1j},\ldots ,x_{nj}\right\}}{max\left\{x_{1j},\ldots ,x_{nj}\right\}-min\left\{x_{1j},\ldots ,x_{nj}\right\}},$$

Negative index:(10)$$x_{ij}^{\prime }=\frac{max\left\{x_{1j},\ldots ,x_{nj}\right\}-x_{ij}}{max\left\{x_{1j},\ldots ,x_{nj}\right\}-min\left\{x_{1j},\ldots ,x_{nj}\right\}},$$

Calculate the contribution or characteristic ratio of the expert i for index j using the following formula.
(11)$$p_{ij}=\frac{x_{ij}^{\prime }}{\sum_{i=1}^{n}x_{ij}},$$

For entropy calculation, calculate the entropy of the jth index.
(12)$$e_{j}=-\frac{1}{lnn}\sum_{i=1}^{n}p_{ij}\ln\left(p_{ij}\right),0\leq e_{j}\leq 1,$$

Calculate the weight of the evaluation index, $w_{j}$, and finally, normalize the index weight.
(13)$$w_{j}=\frac{g_{i}}{\sum_{i=1}^{m}g_{i}},$$

#### 4.3.2. Data Processing

We established that the sore function and the index weight of each evaluation item in the modules is determined by the entropy method. The index weight for the Expert Group was determined through experimental data, as presented in Table 8.

The index weight and score function model for each module is presented in Table 9.

## 5. Result and Discussion

We use Matplotlib (Python 3.9.3) to create block diagrams that display *p*-values for comparing the same parameters in different training modules between the Novice Group and Expert Group, as depicted in Figure 9.

The figure indicates that the Novice Group requires significantly more time to complete two modules and their surgical instrument moving lengths are notably longer than those of the Expert Group. Instrument trajectories, illustrating the disparities between the Novice Group and Expert Group, were generated using Matplotlib [30] and are presented in Figure 10.

Upon the analysis of Table 7 and Figure 9, it becomes evident that a substantial disparity exists between expert and novice medical practitioners. The Expert Group demonstrate significantly shorter task completion times, reduced distances traveled during both left- and right-hand operations, and notably fewer instances of task failure when compared to their novice counterparts. Furthermore, the statistical significance tests conducted on the data yield *p*-values consistently below 0.05, underscoring the robustness and reliability of the data.

Figure 10 provides further insight into the matter, revealing that expert doctors exhibit methodical and predictable motion trajectories while manipulating virtual surgical instruments. These trajectories closely resemble the movements required for peg transfer and soft tissue cutting tasks under real-world surgical conditions. In contrast, novice doctors’ motion trajectories appear less structured, displaying irregular and less predictable patterns.

### 5.1. Peg Transfer

In Figure 9, a comparison of experimental data between the Novice Group and the Expert Group has been conducted for the peg transfer module. The key distinguishing factor between experts and novices in terms of surgical skill is the number of peg drops. This implies that more experienced operators tend to drop fewer pegs. Additionally, experts complete the training module in considerably less time and with shorter instrument movement lengths. Furthermore, in Figure 10, it is apparent that experts’ instrument trajectories are more uniformly distributed. The right trajectory closely resembles the arrangement of columns in Figure 8a. In contrast, novices take more time to complete the training task, and their instrument trajectories lack discernible patterns.

### 5.2. Soft Tissue Cutting

It is evident from Figure 9 that the Novice Group require significantly more time, longer instrument movement lengths, and a higher cutting frequency to complete the soft tissue cutting task compared to the Expert Group. Furthermore, as illustrated in Figure 10, the experts exhibit a higher level of precision in their soft tissue cutting skills, resulting in instrument trajectories that are more concentrated on a fixed path. In contrast, novices tend to exhibit more dispersed instrument trajectories when compared to experts.

### 5.3. Limitation of the VRDT-RMIS

Limited Assessment Metrics: The relatively limited set of assessment metrics used to evaluate the performance of participants is considered one of the limitations. While specific parameters were the focus of this study, future research could benefit from a more comprehensive range of metrics to provide a more nuanced evaluation of surgical skills.

Lack of Post-Training Assessment: A post-training assessment to measure the extent of improvement in the surgical abilities of novice participants after training was not conducted. Such assessments should be considered in future research to gain a comprehensive understanding of the simulator’s training effectiveness.

Small Sample Size: A sample size of 25 participants was involved in our study, which, while sufficient for the initial evaluation, may benefit from larger sample sizes in future studies to enhance the generalizability of the results.

Realism of the Virtual Environment: While aiming to replicate real surgical scenarios, the simulator’s realism can be further improved in certain aspects to better mimic real-world conditions.

In future research, we intend to address these limitations by expanding our assessment metrics, conducting post-training assessments, increasing sample sizes, and refining the realism of the virtual environment. These enhancements will contribute to a more comprehensive and accurate evaluation of the simulator’s effectiveness in surgical skill training.

### 5.4. Comparison with Existing RMIS Simulators

To better demonstrate the improved performance of the VRDT-RMIS simulator, a comparison was conducted with some of the existing surgical robot simulators [31]. It was found that our simulator not only closely approximates the dV-Trainer^®^ (MultiCare Health Systems, Tacoma, WA, USA) in terms of robotic simulation, 3D visualization, and 3D mode, but also includes a high-refresh-rate haptic feedback that is lacking in most simulators. The information mentioned above is presented in Table 10.

## 6. Conclusions

In this paper, we used the RCM task to constrain the motion of the robot digital twin in a Unity3D virtual environment and control it in real time through haptic devices. Then, we set up two surgical skill training modules within the virtual environment and invited 25 volunteers to verify the effectiveness of the simulator. An entropy method was employed to assign weights to each parameter during the experiment for the purpose of comparing the performance of the Novice Group and the Expert Group. Last but not least, we discussed the recorded data and found that the simulator can extremely distinguish novices and experts. However, due to a lack of time, we did not compare the improvements in surgical abilities and the extent of improvement for novices after training. This is something that we will further strengthen in our future work.

Integrating VRDT-RMIS into routine surgical training in modern surgical departments is our long-term vision. To achieve this goal, we will focus on the following improvements:Enhancing the coherence between real robot control and the virtual robot environment in the simulator remains a priority. We aim to seamlessly integrate these two components into a responsive system.Given the significant differences in communication speeds today, remote surgery is becoming a reality. We are committed to advancing the connectivity of the VRDT-RMIS system, allowing surgeons to access it from different locations, facilitating guidance and training from experts worldwide.The simulator’s data collection capabilities are invaluable for objective skill assessment and surgical competency certification. Surgeons can use the system to track their progress, identify areas for improvement, and attain certification based on their proficiency levels.

Furthermore, our objective is to conduct broader clinical validation studies to demonstrate the simulator’s effectiveness in improving real-world surgical outcomes.

## Figures and Tables

**Figure 1 bioengineering-10-01302-f001:**
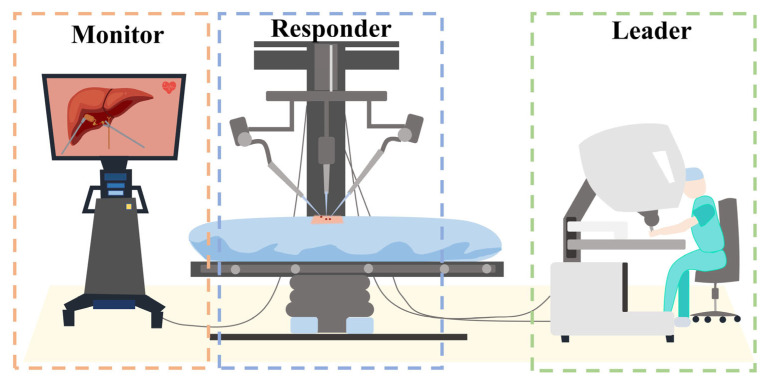
Leader–responder framework: the secondary robot is not directly controlled by the surgeon’s hand but is manipulated by the surgeon through motion capture using the leader console.

**Figure 2 bioengineering-10-01302-f002:**
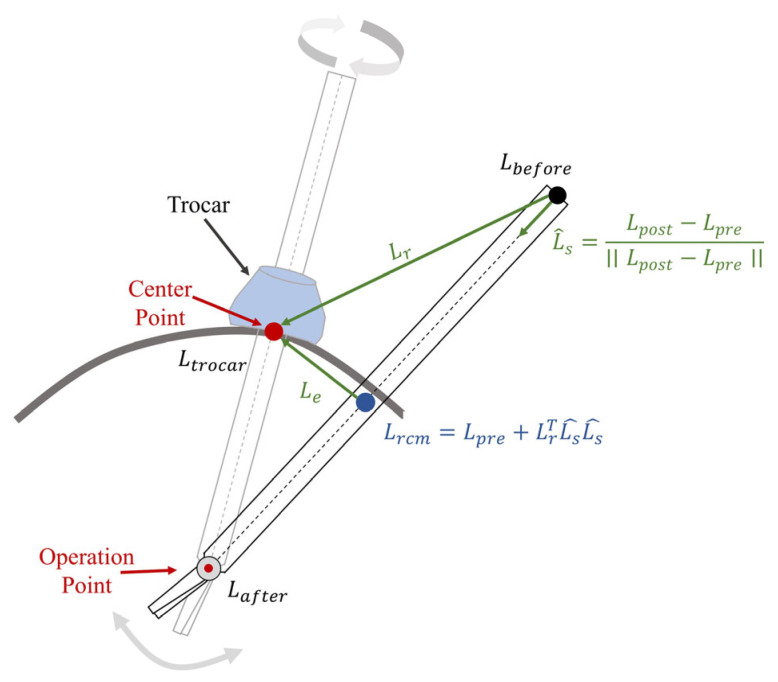
In robotic minimally invasive surgery (RMIS), the remote center of motion (RCM) process restricts the movement of surgical instruments through a trocar point.

**Figure 3 bioengineering-10-01302-f003:**
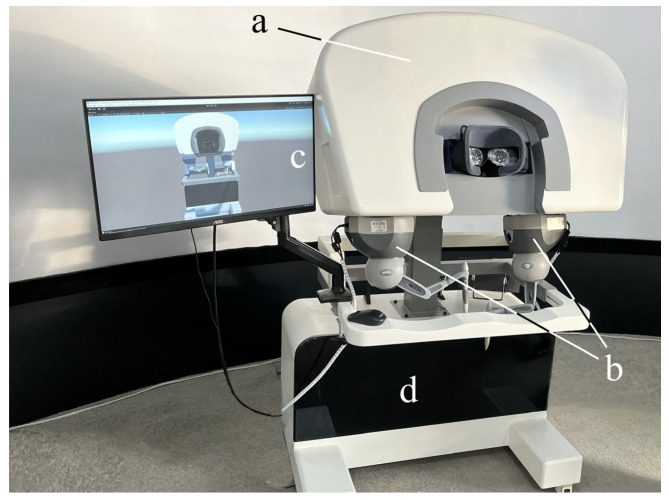
The hardware equipment of the leader part: (a) 200°FOV VR glasses (Pimax^®^, Shanghai, China). (b) Two 3DSystems Touch devices (3D Systems^®^, Rock Hill, SA, USA). (c) Monitor for exhibiting the digital-twin robot (AOC, Fuzhou, Fujian, China). (d) Computing station with Windows 10 (Intel(R) Core(TM) i7-6700K CPU @ 4.00 GHz, Santa Clara, CA, USA) system and an Nvidia GTX3060 graphics card (Nvidia^®^, Santa Clara, CA, USA).

**Figure 4 bioengineering-10-01302-f004:**
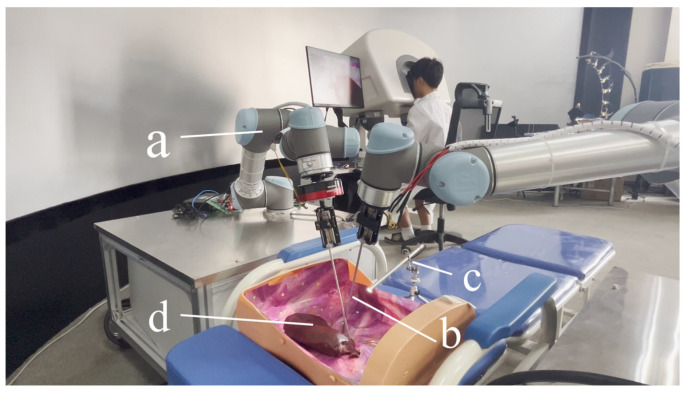
Equipment of the responder part. (a) Two UR5 robots. (b) End effectors to operate target objects. (c) Endoscope with 1920 × 1080 resolution. (d) Target object.

**Figure 5 bioengineering-10-01302-f005:**
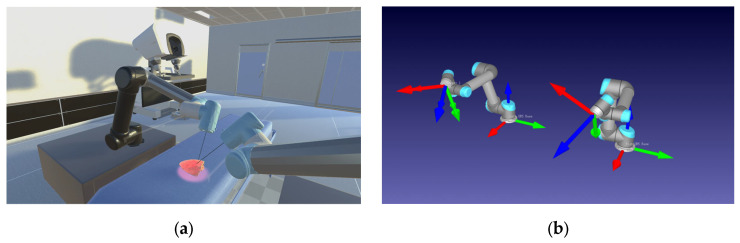
Environment for digital-twin robot control in real time. (**a**) Unity3D vision rendering environment. (**b**) RoboDK digital-twin robots (The arrows represent the Transform Frame (TF).).

**Figure 6 bioengineering-10-01302-f006:**
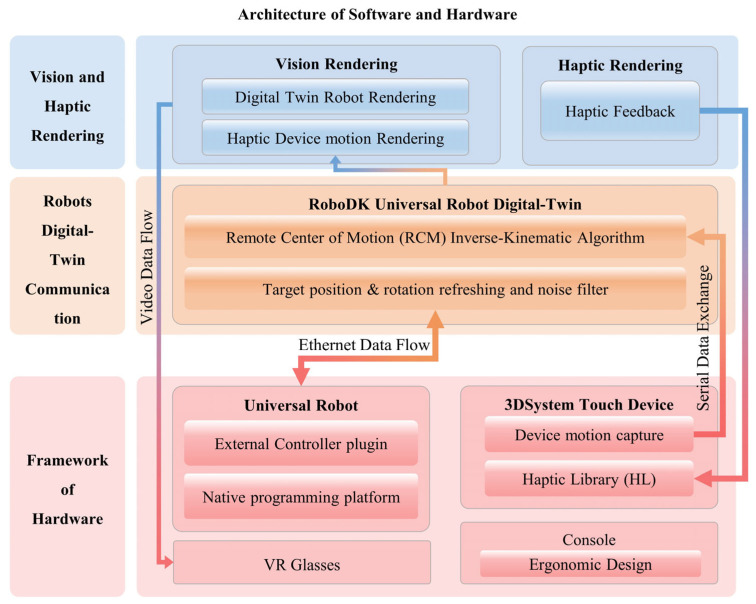
The system’s architecture encompasses both software and hardware components and can be primarily divided into three main sections. Hardware layer: This layer includes a Universal Robot, a 3Dsystem Touch device, and VR glasses. The Touch device’s motion capture plugin transmits data via a serial connection to the RCM IK algorithm, while the UR exchanges data through an ethernet connection with the digital-twin communication layer in RoboDK. Digital-Twin Communication Layer: within this layer, data exchanged by the Universal Robot are processed and managed. Vision and Haptic Rendering Layer: This layer is responsible for rendering video data and haptic feedback, which are subsequently relayed to the VR glasses and Touch device. The entire architecture ensures seamless interaction and communication between the software and hardware components, facilitating the control and feedback loop for the digital-twin robots.

**Figure 7 bioengineering-10-01302-f007:**
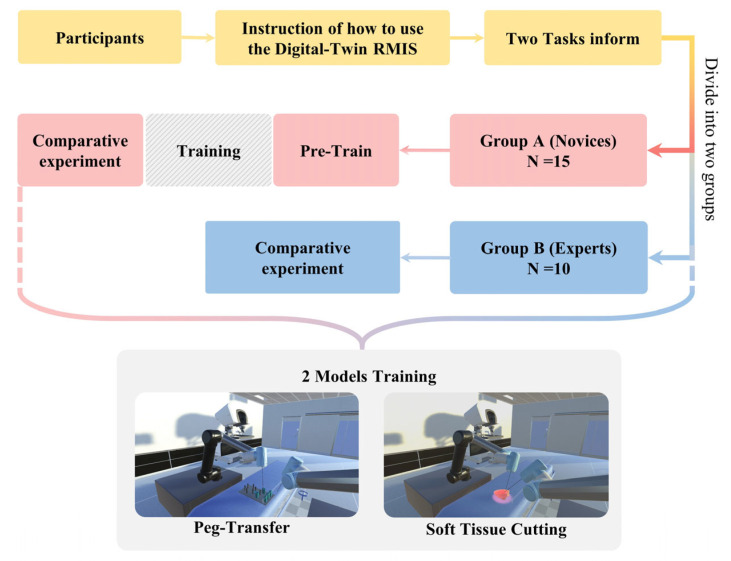
Period of the whole experiment. Participants are divided into two groups since they are instructed about how to use the digital-twin RMIS platform and the tasks of the experiment.

**Figure 8 bioengineering-10-01302-f008:**
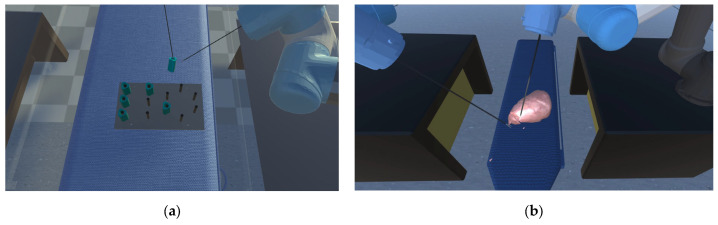
(**a**) Peg transfer module: the operators should employ the left surgical instrument to elevate the small pegs positioned on the left side, and subsequently use the right surgical clamp to grasp and transfer the pegs to the columns on the right. (**b**) Soft tissue cutting module: the operators must utilize the left surgical clamp to grasp and gently tug on the soft tissue, followed by using the right surgical knife to precisely remove the specified soft tissue entirely.

**Figure 9 bioengineering-10-01302-f009:**
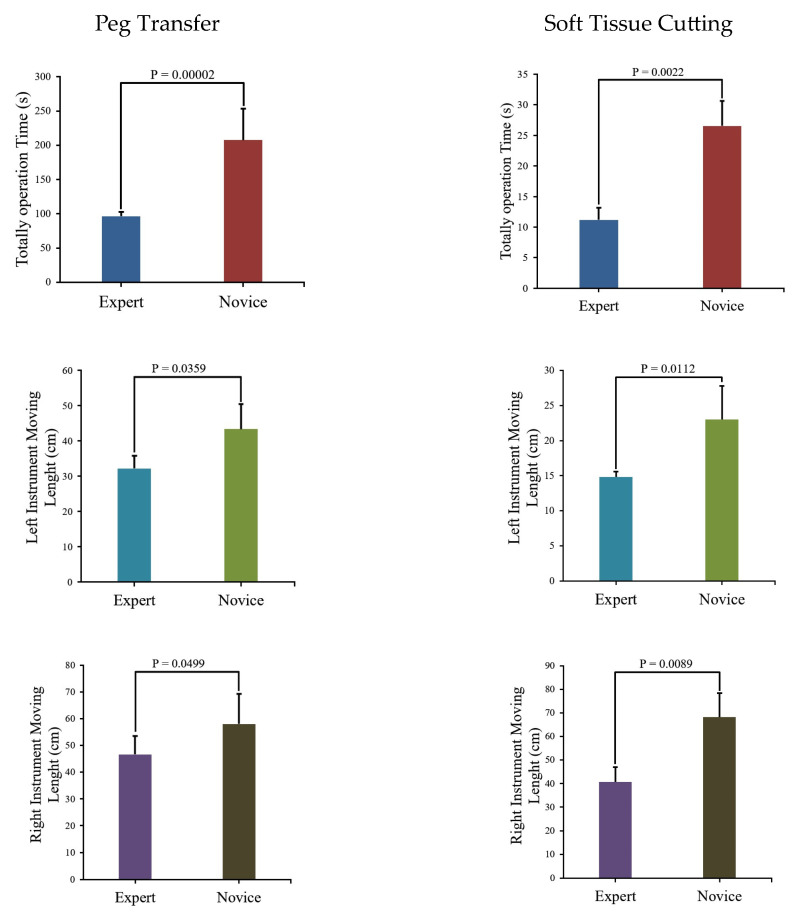
Comparison of module parameters and associated *p*-values between the Novice Group and Expert Group.

**Figure 10 bioengineering-10-01302-f010:**
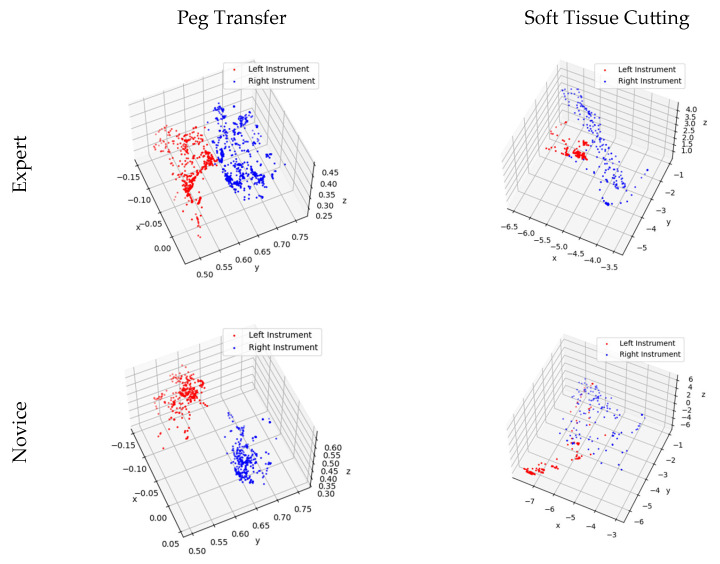
Trajectory of surgical instrument movement in the training modules for the Novice and Expert Groups.

**Table 1 bioengineering-10-01302-t001:** Hardware components of the leader.

Component	Leader Part (Figure 3)
a	VR glasses
b	1000 Hz Haptic Devices
c	Monitor
d	Computing Station

**Table 2 bioengineering-10-01302-t002:** Hardware components of the responder.

Component	Responder Part (Figure 4)
a	UR5 Robot
b	End-effectors
c	Endoscope
d	Target object

**Table 3 bioengineering-10-01302-t003:** Participants in two groups.

Information	Group A (Novice)	Group B (Expert)
Number	15	10
Average age (years)	27.8 (25–30)	46.5 (42–55)
Postgraduate year of training (years)	3 (2–4)	8 (6–10)
Male (%)	66.7	70
Right-handed (%)	86.7	90
Real RMIS experience	8/15	10/10
VR RMIS experience	7/15	5/10

**Table 4 bioengineering-10-01302-t004:** Questionnaire of the face and content validity.

Face and Content Validity Questions (Score: 1–5; 1 = Poor, 5 = Excellent)	
Question 1	Visual Realism of the Model
Question 2	Realism of the Surgical Instrument
Question 3	Realism of Robot Control
Question 4	Haptic Feedback Realism
Question 5	Realism of the Surgical Environment

**Table 5 bioengineering-10-01302-t005:** Subjective questionnaire results of face and content assessment.

	Peg Transfer	Soft Tissue Cutting	Mean
Question 1	4.2	4.4	4.3
Question 2	3.8	4.4	4.1
Question 3	4	4.6	4.3
Question 4	3.8	4	3.9
Question 5	4.2	4.2	4.2

**Table 6 bioengineering-10-01302-t006:** Construct validation of face and content assessment.

	Peg Transfer				Soft Tissue Cutting			
	*T*	*LML*	*RML*	*ND*	*T*	*LML*	*RML*	*FC*
Expert	119.82	34.15	54.34	1	15.80	15.21	37.43	3
Novice	240.22	34.96	77.28	6	26.96	24.34	64.36	6

**Table 7 bioengineering-10-01302-t007:** The comparison of objective parameters for construct validity.

	Peg Transfer				Soft Tissue Cutting			
	*T*	*LML*	*RML*	*ND*	*T*	*LML*	*RML*	*FC*
Expert	0.00002	0.0359	0.0499	0.0007	0.0022	0.0112	0.0089	0.0005
Novice								

**Table 8 bioengineering-10-01302-t008:** Data for each training module experiment (experts).

	Peg Transfer				Soft Tissue Cutting			
	*T*	*LML*	*RML*	*ND*	*T*	*LML*	*RML*	*FC*
Expert1	91.00	35.70	53.54	1	14.11	14.46	47.17	3
Expert2	92.18	26.28	37.83	0	8.61	15.95	34.55	3
Expert3	105.34	32.75	52.02	1	10.60	14.24	34.46	2
Expert4	90.28	32.00	48.72	0	11.64	15.25	40.94	3
Expert5	93.34	34.22	40.85	0	10.98	14.11	46.76	3
Expert6	107.32	33.98	53.89	1	12.65	15.41	39.73	3
Expert7	92.61	30.85	45.31	0	9.73	14.36	35.67	2
Expert8	96.52	31.50	51.49	1	12.56	16.91	45.33	3
Expert9	102	31.84	54.71	0	13.11	15.26	42.09	3
Expert10	92.09	29.81	38.72	0	9.18	14.63	37.95	2

**Table 9 bioengineering-10-01302-t009:** Weight and score function model of metrics in each training module.

	Peg Transfer				Soft Tissue Cutting			
	*T*	*LML*	*RML*	*ND*	*T*	*LML*	*RML*	*FC*
Weight	0.50	0.43	0.44	−0.36	0.20	0.28	0.29	0.23
Score	S=0.5×T+0.43×LML+0.44×RML−0.36×ND				S=0.20×T+0.28×LML+0.29×RML+0.23×FC			

**Table 10 bioengineering-10-01302-t010:** Comparison with the existing RMIS simulators.

Simulators	Country	Robotic Simulation	3D Version	3D Mode	Haptic Feedback
dV-Trainer^®^	Tacoma, WA USA	Yes	Yes	Fingertip Operation in Closed Binoculars	No
HUGO™ RAS	Minneapolis, MN USA	No	Yes	Open 3D Glasses, Laparoscopic Handle	Yes/960 Hz
MedBot^®^ Toumai	Shanghai CHN	No	Yes	Open 3D Glasses, Laparoscopic Handle	No
VRDT-RMIS	Kunming, Yunnan CHN	Yes	Yes	Fingertip Operation in Closed Binoculars	Yes/1000 Hz

## Data Availability

The data presented in this study are available on request from the author Xiaoyu Cai.

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
