# Peer review of "Implementation of a Virtual Reality Based Digital-Twin Robotic Minimally Invasive Surgery Simulator"

_bioengineering, 2023, doi:10.3390/bioengineering10111302_

Round 1

Reviewer 1 Report

Comments and Suggestions for Authors

This manuscript aims to determine the effectiveness of simulator training. Not being a robotic expert, but a clinician with training in epidemiology, it's difficult for me the determine the validity and rigor of the proposed method to assess the effectiveness of the simulator.

The topic of the manuscript is interesting. The manuscript is very disorganized. Before a recommendation for publication, the content of the different sections has to be properly structured: for instance, methods are described in the Results, which makes it very difficult to follow.

Tables are complex to understand, as they don't have a common logic (Table 3 describes only results for experts, but not novices). Figures appear to overlap what is describe what is shown in Tables.

Reviewer 2 Report

Comments and Suggestions for Authors

Dear Authors;

Here is a review based on the paper:

This paper presents a novel Virtual Reality-based Digital Twin Robotic Minimally Invasive Surgery (VRDT-RMIS) simulator for surgical training. As robotic surgery becomes more commonplace, there is a need for effective training tools to help surgeons develop the specialized skills required. The VRDT-RMIS simulator aims to provide an immersive virtual environment with haptic feedback for practicing robotic surgery maneuvers. 

The authors developed two training modules focused on peg transfer and soft tissue cutting skills. 25 volunteers were recruited to test the simulator, including experienced robotic surgeons (Expert Group) and medical students (Novice Group). The simulator was able to effectively distinguish between the skill levels of the two groups based on performance metrics in the modules. This demonstrates the simulator's validity for surgical skills assessment.

A particular strength of this research is the use of an entropy-based weighting system to objectively analyze the performance data. This allowed comprehensive comparisons between novices and experts over multiple parameters. The immersive VR environment also provides key visual and haptic feedback to enhance the realism.

While results are promising, the study is somewhat limited by the small sample size of participants. Additionally, the long-term training effects were not evaluated. Further research on skill acquisition curves over multiple training sessions would provide useful insights. Overall, this is an important step forward in VR-based surgical simulation with significant potential to improve robotic surgery training. The authors have developed an innovative platform that warrants expanded research and development.

Here are some suggestions to improve the review:

- Expand more on how the VRDT-RMIS simulator works and its key features. Provide additional details on the virtual environment, haptic feedback system, and training modules.

- Give more specifics on the study methodology - how many participants were in each group, what surgical tasks they performed, and what metrics were used to evaluate them.

- Provide more details on the results - what was the extent of the performance difference between novices and experts? How well could the system differentiate their skills?

- Discuss any limitations of the current simulator system or study methodology and how they could be improved in future research.

- Elaborate on the implications/applications of the research - how could this simulator be incorporated into surgical training programs? What are the next steps for developing it further?

- Comment on how this compares to other existing surgical simulation systems and virtual reality trainers. Does it provide any advantages over current methods?

- Suggest any follow-up studies that could be done to further validate the effectiveness - such as testing skill acquisition over multiple training sessions.

- If possible, provide additional insights into the entropy-based weighting system and how it enabled objective performance analysis.

- Expand the conclusion to summarize the key findings and highlight the potential of VR-based surgical simulation. Reiterate the future research needed in this area.

Reviewer 3 Report

Comments and Suggestions for Authors

I am a surgeon with a lot of practical expertise in the field but some physics formulas are too complicated for me. It seems an interesting paper, as far as I can assess.
